# Multimodal neural networks better explain multivoxel patterns in the hippocampus

**Bhavin Choksi**
CerCo CNRS, UMR 5549 &
Université de Toulouse
bhavin.choksi@cnrs.fr

**Milad Mozafari**
CerCo CNRS, UMR 5549 &
IRIT CNRS, UMR 5505
milad.mozafari@cnrs.fr

**Rufin VanRullen**
CerCo CNRS, UMR 5549 &
ANITI, Université de Toulouse
rufin.vanrullen@cnrs.fr

**Leila Reddy**
CerCo CNRS, UMR 5549 &
ANITI, Université de Toulouse
leila.reddy@cnrs.fr

## Abstract

The human hippocampus possesses "concept cells", neurons that fire when presented with stimuli belonging to a specific concept, regardless of the modality. Recently, similar concept cells were discovered in a multimodal network called CLIP [1]. Here, we ask whether CLIP can explain the fMRI activity of the human hippocampus better than a purely visual (or linguistic) model. We extend our analysis to a range of publicly available uni- and multi-modal models. We demonstrate that "multimodality" stands out as a key component when assessing the ability of a network to explain the multivoxel activity in the hippocampus.

## 1   Introduction

The field of machine learning has experienced tremendous breakthroughs in the past few years. A hallmark of these breakthroughs are deep neural networks (DNNs) that can solve complex tasks going beyond computer vision to tasks requiring semantic knowledge and understanding, features characteristic of human intelligence (e.g., story completions, context-based question answering, code generation etc.,). This feat has been made possible by both the ability of DNNs to learn expressive representational spaces that enable them to carry out these complex tasks, as well as by the development of improved optimization algorithms required to train the DNNs.

Importantly for the neuroscience community, DNNs also provide a potential model for understanding the human brain. Their mathematical pliability combined with their unprecedented expressivity has opened up novel avenues to investigate the human brain. Efforts are being made to understand the similarities and differences between these two systems due to their architectures, dynamics, behavioral patterns, and representational structures [2–5].

At the same time, DNNs themselves are getting better and better on more human-like tasks. Recently, Radford et al. [1] proposed a model that could simultaneously learn visual and linguistic information from a huge dataset using a constrastive loss function. Importantly, this multimodal model, known as CLIP, was found to possess neurons in its last layer that encoded specific concepts [6]. These artificial neurons are reminiscent of 'concept cells' in the human medial temporal lobe (MTL) [7, 8], biological neurons that appear to represent the meaning of a given stimulus or concept in a manner that is invariant to how that stimulus is actually experienced by the observer. For example, a single neuron in the human hippocampus showed incredible specificity in its response

3rd Workshop on Shared Visual Representations in Human and Machine Intelligence (SVRHM 2021) of the Neural Information Processing Systems (NeurIPS) conference, Virtual.

to the actress Halle Berry. This neuron responded to different images of the actress, including to photographs in which she was disguised as Catwoman (her starring role in a movie by the same name). The same neuron also responded to a semantic representation of the concept, i.e. to the letter string "HALLE BERRY". Other studies have since shown that "concept cells" are also activated when stimulus information is provided in other sensory modalities, for example when the name of the person is spoken out loud [9].

The discovery of concept cells in artificial networks raises a natural question — *Can a multimodal model like CLIP explain the activity of brain regions known to possess concept cells better than a purely visual model?* In this work we investigate this question by using publicly available fMRI data [10], and asking if CLIP can explain the activity of the hippocampus region better than a comparable feedforward visual model, i.e., ResNet. Because fMRI data does not provide us with the spatial resolution to identify individual concept cells, we address this question at the level of multi-dimensional representation spaces rather than at the level of individual neurons. We also extend our analysis to a variety of models from the literature, trained with unimodal or multimodal objectives. Using Representational Similarity Analysis (RSA) [11], we report that multimodal networks consistently rank higher than their unimodal counterparts in their ability to explain fMRI activity in the human hippocampus. We provide all the code for reproducing our results, and easily extending the analysis to other models in the future, on Github[1].

## 2 Methods

### 2.1 RSA

Representational Similarity Analysis (RSA) [11] compares stimulus representations across different high-dimensional spaces (e.g., brain multi-voxel spaces, model latent spaces, etc.). A first step in RSA consists of constructing Representational Dissimilarity Matrices (RDMs) in each space. RDMs are two-dimensional matrices, in which each element measures the pairwise distance between two stimulus conditions. An important property of RDMs is that their size is the same regardless of the initial dimensionality of each space, since the number of elements in an RDM only depends on the number of conditions being compared. RDMs from representational spaces of different dimensionalities can thus be compared to each other. Since the number of elements in an RDM only depends on the number of conditions being compared, RDMs from representational spaces of different dimensionalities can be compared to each other. In this work, we use the Pearson correlation distance (defined as 1 - correlation) to construct the RDMs, and subsequently compare them with the Pearson's $r$ correlation coefficient. Results with other choices of metrics are shown in the Appendix.

### 2.2 fMRI data

For our investigations, we use publicly available data from [10]. This dataset consists of fMRI data collected on five healthy participants viewing images from a subset of categories available in ImageNet. Participants performed a one-back test in the scanner in which they had to press a button when the same image was repeated on two consecutive presentations. The data were collected on 1200 training images that were presented once, and 50 test images presented 35 times each. For our experiments, we restrict ourselves to the subset of test images since the higher number of repetitions provides a more robust estimate of the multi-voxel representation of each image.

After downloading the raw fMRI data, we preprocessed them with a standard pipeline using SPM12 [12]: slice-time correction, realignment, and coregistration to the T1W (T1-Weighted) anatomical images. We performed a general linear model (GLM) using regressors for each image (the onset and duration), along with regressors for 'fixation' and 'one-back'. For each subject, the beta coefficients obtained from the GLM were transformed into a common MNI305 space [13] using FreeSurfer ([2]) to allow analysis across subjects. We defined four regions of interest (ROIs) using the Deskian-Killiany atlas [14] for both the left and right hemispheres: a visual region comprising the lateraloccipital and pericalcarine regions, a fusiform region, a hippocampal, and a parahippocampal ROI. fMRI RDMs were built using the beta values in each ROI for each subject. Since 50 image conditions were compared, each RDM was 50x50 in squareform.

---

[1]Code available at: `https://github.com/bhavinc/mutlimodal-concepts`

[2]`http://surfer.nmr.mgh.harvard.edu`

## 2.3 Models

We include a variety of models in our analysis to facilitate interpretation and discovery of underlying trends in different classes of models. All the included models are publicly available, and possess a ResNet50 backbone to minimize architectural differences.

For CLIP, we used the visual CLIP-RN50 backbone (called CLIP hereafter), which was jointly trained along with a linguistic head (called CLIP-L hereafter) on a contrastive learning task on 400M image-caption pairs [1]. Additionally we also considered visual features from TSM [15], another multimodal model that is trained with a contrastive objective on the HowTo100M dataset [16], in a task that comprises three modalities (video, text, and audio). The impact of contrastive learning objectives on the features of these models can be compared to VirTex and ICMLM, multimodal networks trained with different objectives. For Virtex, the visual backbone is trained on an image captioning task [17], while for ICMLM, the visual features are trained on a text-unmasking task [18]. Both VirTex and ICMLM are trained on MS-COCO [19], a much smaller dataset compared to those used for CLIP or TSM.

To tease apart the effect of multimodal training, we also included visual-only models in our comparisons. Since dataset size has been suggested to affect the quality of representations learned by a network, we considered two visual-only models trained on different datasets. We used the standard ResNet50 model (the control visual model) trained on ImageNet-1K, as well as BiT-M, a ResNet50 backbone trained on the significantly larger ImageNet-21K dataset [20]. We also included adversarially robust models (AR-L2, AR-L4, AR-L8) from [21] in our comparisons. These models are trained to be robust to minute perturbations to the input images by explicitly incorporating such perturbed (adversarial) images [22] in the training dataset. These models have been observed to possess more human-like features compared to standard feedforward versions [23], making them particularly relevant to our analysis.

Unlike human observers who rely on shapes, standard ImageNet models are strongly biased by the texture of images [24]. Therefore, Geirhos et al. [24] designed a stylized version of ImageNet to train models that have a stronger bias towards shape than texture. To assess whether representations optimized for human-like biases are better at explaining brain activity in MTL regions, we included three StylizedImageNet models in our comparisons: (i) a model pretrained on only StylizedImageNet (SIN) images, (ii) a model trained on SIN images and ImageNet combined (SIN-IN), and (iii) the SIN-IN model further fine-tuned on ImageNet (SIN-IN+FIN).

Finally, apart from visual and multimodal models, we also included language models: GPT-2 [25], BERT [26], as well as CLIP-L. Although these models are not trained to process visual data, they provide an important basis for comparison along with visual and multimodal networks.

For multimodal and visual backbones, we used the test images shown to the human participants and obtained their feature representations from the final average pooling layer. For language models, for each image, we encoded the text 'a photo of {ImageNet label of the image}' to obtain the latent representations. For each model, these latent representations were then used to obtain the RDMs of shape 50x50.

## 2.4 Voxel Selection from anatomical ROIs based on Noise ceilings

We start by evaluating the signal of the selected beta coefficients. In each ROI, we calculated the noise-ceiling, defined as the average inter-subject correlation between RDMs. The noise ceiling provides an estimate of the reliability of the fMRI signal in a given ROI across subjects. Due to the visual nature of the task, the more visual regions (visual ROI, fusiform and parahippocampus) unsurprisingly showed higher values for the noise-ceiling (between 0.2 and 0.6). In contrast, the noise ceiling in the hippocampus was relatively low, and not significantly different from zero ($-0.012 \pm 0.012$). This could be due, in part, to the fact that the fMRI signal in the hippocampus is generally less reliable. However, single neuron recordings in the human hippocampus have also revealed that only a small proportion of cells ($\approx 15\%$ of recorded cells) is responsive to visual stimuli, and even fewer ($\approx 5\%$) qualify as "concept cells". In fact, the hippocampus is well-known for its implication in non-visual tasks, i.e. spatial navigation or memory retrieval and consolidation. If only a small subset of voxels in the hippocampus respond to visual stimuli, it stands to reason that a noise ceiling computed across all voxels would not capture any meaningful visual information.

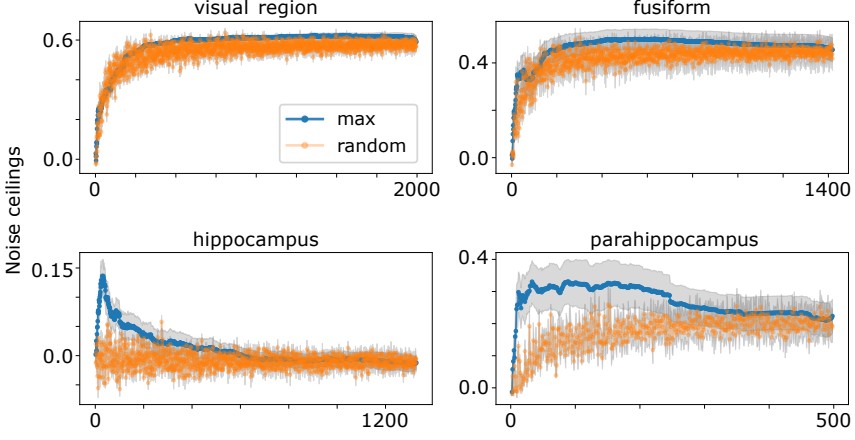

Figure 1: **Noise ceilings after selecting subsets of voxels from each region** The panels show the noise ceilings (i.e., inter-subject correlation) calculated after selecting different numbers of voxels from each region of interest. The noise ceilings were computed using either voxels with the highest beta values (blue) or via a random sampling of voxels (orange). The gray regions denote the standard error of mean. For certain ROIs (visual region, fusiform), most voxels are informative about the visual stimulus, and the two selection methods yield similar results. For other ROIs (hippocampus, parahippocampus), the noise ceiling depends on the selection method, implying that some voxels (with the highest betas) are more informative than others (randomly selected). The hippocampus shows an improved noise ceiling when 30 voxels with the highest beta values are selected, with additional voxels degrading the signal.

To circumvent this issue, we defined a quantifiable criterion to select a limited number of voxels from each ROI. Specifically, we selected the $N$ voxels with the highest beta value (for any of the 50 stimuli), and calculated the noise ceiling based on this voxel selection. We varied $N$ systematically. As a control, we used random selections of $N$ voxels. As Figure 1 shows, the noise ceiling in the more visual regions (visual region, fusiform, parahippocampus) increased rapidly and then stabilized after the inclusion of ≈20% of the total voxels. This was true, even when the voxels were randomly selected, indicating that most voxels in these regions carry information about visual stimuli. In the hippocampus however, the noise ceiling was virtually zero when based on random voxel selections: most hippocampal voxels do not appear to encode visual information. Nonetheless, when selecting the $N$ most-activated voxels, the noise ceiling peaked at ≈30 voxels, before sharply dropping down to random levels. This is consistent with our hypothesis that although a relatively small number of hippocampal voxels are reliably activated by visual inputs, the signal in these voxels (as measured by the noise ceiling) is reliably above chance. For the main RSA analysis, we thus considered only these top-30 hippocampus voxels. Note that this selection criterion only ensures that the considered brain responses are meaningful, but does not bias the outcome of the RSA with neural network models (i.e., there is no danger of circular reasoning). For the other ROIs, we also considered the top-30 voxels for a fair comparison; yet we also report a different selection procedure (based on a fixed beta threshold) in the Supplementary Material.

## 3  Results

To investigate whether CLIP explains multivoxel activity patterns in MTL regions better than its visual (or linguistic) counterparts, we computed RSA between the brain RDMs and each model RDM. The noise ceiling places an upper limit on brain-model comparisons because it is an estimate of inter-subject variability. Thus, we normalized the RSA values by the noise-ceiling to allow for comparisons across models and across regions. The normalized RSA values for each model in each ROI are shown in Figure 2A, and averaged across groups of models in Figure 2B. (The corresponding non-normalized values are shown in the Appendix.)

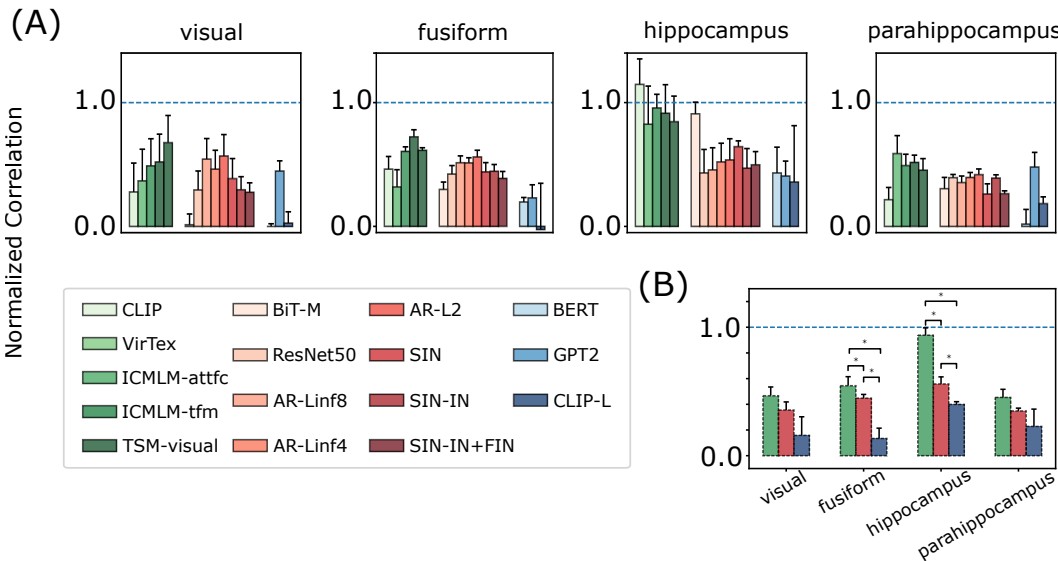

Figure 2: **Multimodal models better explain fMRI response patterns in the hippocampus**: Panel A shows the correlation values obtained with different models across selected regions of interest (ROI). Only 30 voxels were selected from each ROI. The values are normalized with the noise-ceilings to facilitate comparisons across regions. Panel B shows the correlation values after aggregating them over mutlimodal (green), visual (red) and language (blue) models. Statistical significance is calculated by using Welch's t-test and is denoted by an asterisk.

RSA values for the majority of models and brain regions were positive. However, comparisons between individual models (e.g., CLIP vs. ResNet in hippocampus) were not significant (Wilcoxon signed-rank test, p<0.05), possibly because of the small number of fMRI participants. Thus, we grouped the models according to their modalities (for example, BERT, GPT2 and CLIP-L as the *language models*). We thus obtained three classes of models (visual, language, multi-modal), and asked whether one class outperformed the others in explaining brain activity in each ROI. In the hippocampus, in line with our main hypothesis, multimodal models significantly better explained activity patterns compared to both visual and linguistic models (Welch's t-test, p < 0.05. Figure 2B). In fact, the multimodal networks reached the noise ceiling in the hippocampus, meaning that they could explain all of the explainable variance in brain responses–this result did not happen for any other model group in any other ROI. A similar trend was observed in other regions (even reaching statistical significance in the fusiform ROI), but the RSA values were lower and more variable compared to the hippocampus. Finally, the visual and vision-language models performed systematically better than the linguistic models–as expected since all stimuli were visual.

In our main analysis, the RSA was performed using a subset of 30 voxels that showed the highest beta values in each ROI. While this threshold is reasonable in the hippocampus based on our noise-ceiling calculations (Fig 1), visual regions did have a larger number of voxels with reliable beta values. Thus, in a control analysis, in each ROI we selected the $N$ voxels that had beta values greater than a common threshold (determined so as to yield 30 hippocampal voxels). Figure 4 in the appendix shows that including a larger number of voxels had little impact on the main results shown in Figure 2. Finally, in the Appendix, we confirmed that the trends observed in Figure 2 are robust to the choice of distance metrics by using other metrics commonly used for fMRI data.

## 4   Discussion and Conclusion

We applied RSA to study the ability of different neural network models – multi- or uni-modal – to explain the fMRI activity patterns in various brain regions. Based on recent findings [6], our hypothesis was that CLIP (and similar multimodal networks) would be specifically adept at explaining brain activity in the hippocampus–where 'concept cells' are found. This hypothesis was supported by the data: the *multimodal* nature of a model was as a key component in explaining the activity in the

human hippocampus – a trend that proved robust to different methods of voxel selection and distance metrics.

Our findings could provide a novel insight for making more brain-like models. While many studies have reported a reliable correspondence between deep neural network representations and neural activity along the ventral visual pathway, the similarity is far from absolute [27]. In particular, Xu and Vaziri-Pashkam [28] casted doubt on the utility of DNNs for explaining higher regions in the brain. Our findings provide a potential way forward to address this limitation: building models that explain higher regions in the brain might require using datasets spanning different modalities. This can be further combined with bio-plausible architectural changes to the DNNs. For example, it would be interesting to investigate the effects of training a bio-inspired recurrent neural network [29] using multimodal objectives. Combining these architecture- and objective-based approaches could potentially have synergistic effects in learning human-like representations.

We hope the findings in this work further contribute to the efforts in bridging the gap between machine learning and neuroscience.

## Acknowledgments and Disclosure of Funding

RV is funded by ANR grant OSCI-DEEP (ANR-19-NEUC-0004). LR and RV are both funded by an ANITI (Artificial and Natural Intelligence Toulouse Institute) Research Chair (ANR grant ANR-19-PI3A-0004), as well as ANR grant AI-REPS (ANR-18-CE37-0007-01).

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

# A Appendix

## A.1 Non-normalized data corresponding to Figure 1

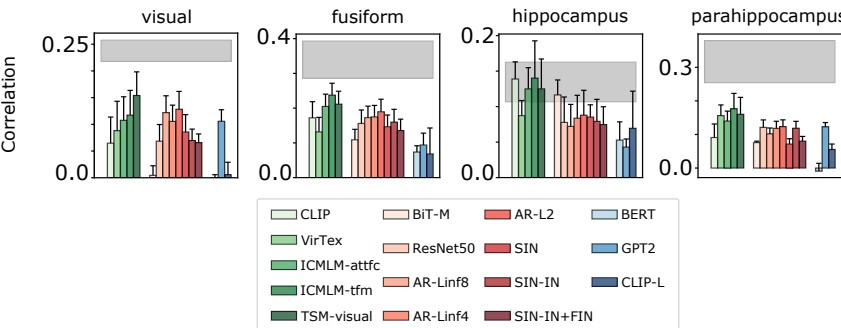

Figure 3: **Non-normalized RSA values between model and brain RDMs.** The brain RDMs are calculated based on selecting 30 voxels from each ROI, as in the main analysis. The gray bands show the upper and lower bounds of the noise-ceilings calculated by adding and subtracting it's s.e.m. respectively.

## A.2 Voxel-selection based on a fixed beta-value threshold.

In the main analysis, we selected 30 voxels in each ROI based on the noise-ceiling analysis in the hippocampus. In other words, in each ROI we selected the 30 voxels with the highest beta values.

As a control method, instead of restricting the number of voxels to 30, we used the value of the 30th voxel from hippocampus as a threshold for other ROIs. The number of voxels found in each ROI for each participant is depicted in Table 1 and the RSA values in Figure 4. We observed that this alternate criterion did not affect the overall trend in other regions.

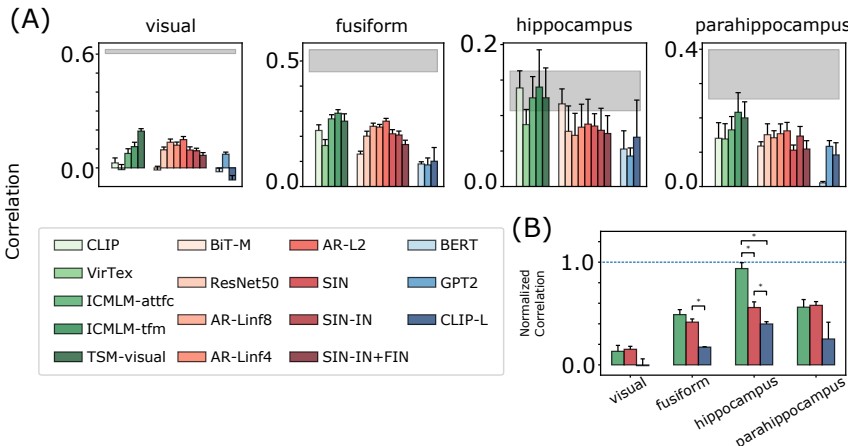

Figure 4: Non-normalized RSA values after using the beta value of the 30th voxel from hippocampus as a threshold for other ROIs for each participant.

Table 1: Number of voxels found in each region after thresholding

|                 | Subject 1 | Subject 2 | Subject 3 | Subject 4 | Subject 5 |
|-----------------|-----------|-----------|-----------|-----------|-----------|
| visual          | 530       | 831       | 343       | 707       | 592       |
| fusiform        | 532       | 376       | 217       | 368       | 508       |
| hippocampus     | 30        | 30        | 30        | 30        | 30        |
| parahippocampus | 122       | 67        | 85        | 111       | 167       |

## A.3 RSA computed using different metrics

We verified the robustness of our results by using other metrics to compute the RDMs and RSA.

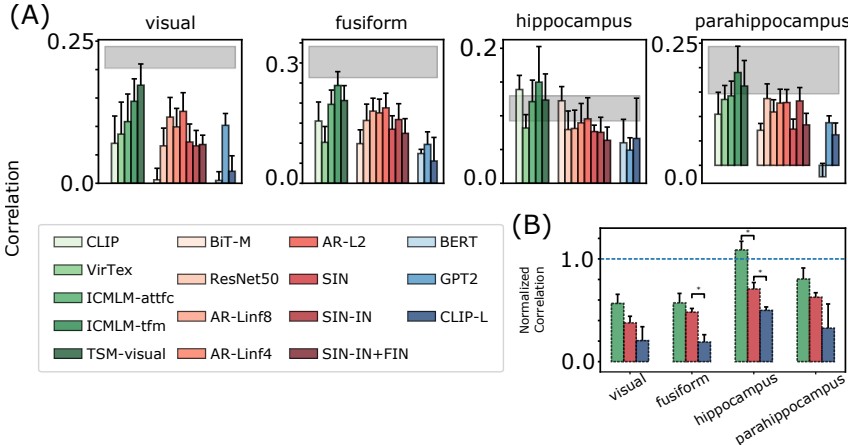

Figure 5: The RDMs were calculated using the Pearson correlation distance, and the Spearman rank correlation was used to compute the RSA.

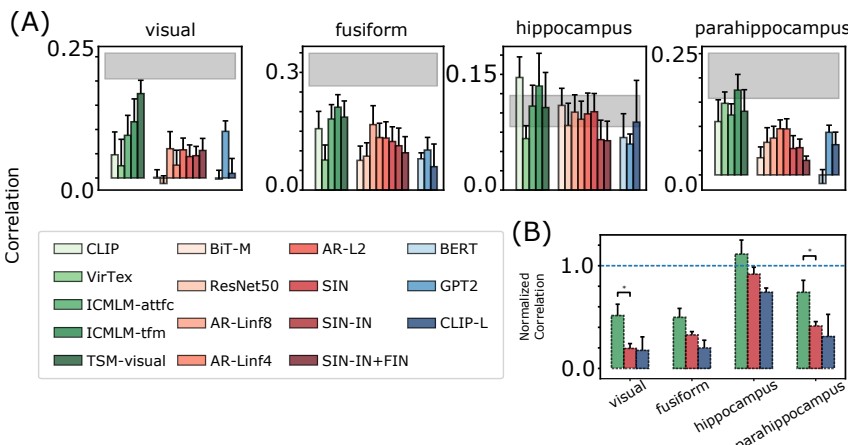

Figure 6: The RDMs were calculated using the Cosine distance, and the Spearman rank correlation was used to compute the RSA.

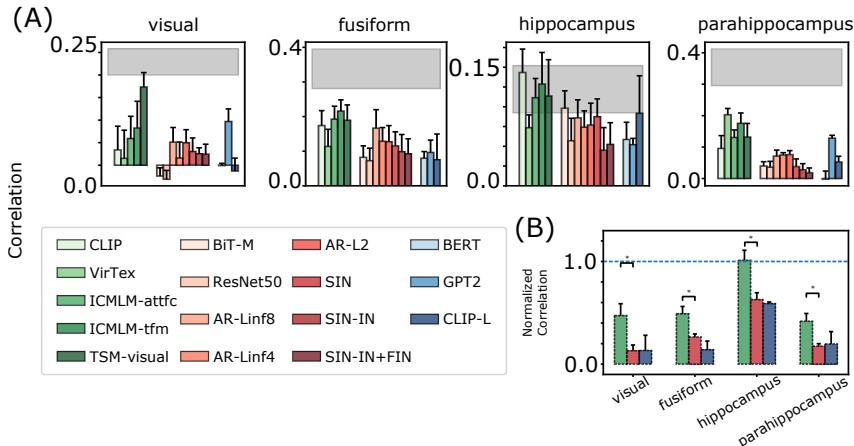

Figure 7: The RDMs were calculated using the Cosine distance, and the Pearson correlation was used to compute the RSA.

## A.4 Licenses of the assets used

| Asset | License |
|---|---|
| FreeSurfer | https://surfer.nmr.mgh.harvard.edu/fswiki/FreeSurferWiki |
| SPM12 | GNU GPL |
| fMRI data | CC0 |
| CLIP | MIT |
| VirTex | MIT |
| TSM | Apache-2.0 |
| ICMLM | N/A |
| BiT-M | Apache-2.0 |
| ResNet | MIT |
| AR models | MIT |
| SIN models | https://github.com/rgeirhos/texture-vs-shape/blob/master/DATASET_LICENSE |
| GPT2 | MIT |
| BERT | Apache-2.0 |

Table 2: Available Licences of all the assets used in the study. Links to the appropriate webpages are provided for special licenses.

## A.5 Broader Impacts

The research discussed above analyses the ability of neural networks to explain the human neural activity, specifically it demonstrates that multimodal neural networks are better than visual or linguistic models in explaining the activity in the hippocampus.

Importantly, this research provides potential insights for designing better bio-plausible networks. Upon diligent use, such networks can elucidate mechanisms in biological brains necessary to help patients. At the same time, we are aware of the possibilities for nefarious use of such systems, and urge all researchers to consider their implications.

