# OpenReview forum: "Multimodal neural networks better explain multivoxel patterns in the hippocampus"
_NeurIPS.cc/2021/Workshop/SVRHM — SVRHM 2021 Oral_

### Official Review · Reviewer_41P3 · 2021-10-23
**Clear, precise and well written paper. Use Representational Similarity Analysis to show that multi-modal DNNs learn representations that are more similar to representations in the Hippocampus compared to uni-modal DNNs.**

**Rating:** 8
**Confidence:** 4

**Review:**

The authors review several DNN models and compare their internal representations with fMRI activity of different brain areas using  Representational Similarity Analysis. In particular, they are interested in evaluating how well representations in new multi-modal DNNs match up with signals in the Hippocampus as both have been shown to include 'concept' cells; cells that respond  to a concept regardless of input type (images, text, audio etc.)

Overall the paper is well written. The data, their methodology  and their results are clearly explained. The scope of the work is small but it is well executed.

Comments:

[1] Many models where used in the analysis. Although all are referenced, the authors do not specify where the code for these models is from. Did they implement them themselves or get them from somewhere?  How did they guarantee they work correctly?

[2] In Figure 2, it is intriguing that multi-modal networks show better alignment with fMRI data from visual areas compared to models trained only on images. Which visual cortices does the fMRI data correspond to? Does it include more than just visual areas? Can the authors expand on this.

[3] Line 58, How do results using the Pearson's correlation coefficient compare to the results obtained by using the other metric?

[4] Lines 67-76, there are a lot of abbreviations that are not explained and are hard to follow.

[5] The example on line 35 needs a reference.

---

### Official Review · Reviewer_3uBV · 2021-10-30
**Interesting analysis on the correlates between multimodal models and brain activity**

**Rating:** 8
**Confidence:** 2

**Review:**

# Summary

Here, the authors tackle the question of whether multimodal models can explain brain activity in regions where it is known to be concept cells better than unimodal models (visual models in particular). They use a publicly available fMRI dataset and via representational similarity analysis, they compare the brain activity and models.

## Strengths

- The paper poses a very interesting question on a relevant topic
- The variety of the SOTA models considered. Not only visual models were included in the comparison, but also language models that were not trained in a visual task but were adapted for it.
- The description and discussion regarding the voxels selection and the noise-celling calculation for each particular ROI
- The clarity of the paper in general

## Weaknesses

- Given the variety of models considered it would be good to have more details on how the latent representations for each model are built
- Results of the statistical tests are not informed, only the type of test performed and whether it was significant or not
- Limitations to compare between specific models

Overall, I find this article highly relevant for the SVRHM workshop. It provides valuable insights on the correlates of multimodal models and brain activity, helping to bring machine learning and neuroscience closer together.

---

### Official Review · Reviewer_xKD9 · 2021-10-31
**Excellent work demonstrating multimodality in hippocampus activity using computational models**

**Rating:** 8
**Confidence:** 5

**Review:**

In this paper, the authors show that the multimodal models better explain fMRI responses in the hippocampus. They use a publicly available fMRI dataset and select a subset of voxels in multiple ROIs based on beta values. Responses in the visual, fusiform, hippocampus, and parahippocampus are then compared to visual, language, and multimodal DNN models.  The results show that in fusiform and hippocampus multimodal models explain responses better than visual and language models.

### Quality
The paper is of high quality due to the following reasons:
1. Clear formulation of the hypothesis and suitable controls within the multimodal models as well as language and visual models
2. Clever use of already existing publicly available datasets and models to validate the hypothesis
3. Suitable statistical tests
4. Robustness of results with respect to the number of voxels and choice of distance metrics.

While overall the results validate the key hypothesis of the paper, there are several other interesting results that might need more discussion. For example, in figure 2A in hippocampus BIT-M seems to be significantly better than other visual models but it shows almost 0 correlation with visual voxels. Why this would be the case? BiT-M is trained on a larger dataset and has been shown to learn more generalizable representation, so it is a bit unclear why it would show almost zero correlation with the visual voxels.

I would also encourage authors to look at layers other than average pooling for a more thorough comparison across different layers of the models and see where the differences start emerging.

### Clarity
The paper is written clearly and the text is easy to follow.

### Originality
The idea of applying multimodal models to explain responses in the hippocampus is novel and has not been done before. This paper also shows how existing datasets and models can be exploited to test new hypotheses.

### Significance
The overall findings are significant for the community interested in applying the computational models to test new hypotheses about the functions of the human brain. The study clearly demonstrates the potential of this approach and thus can be extended beyond the hippocampus to other brain areas.

---

### Decision · Program_Chairs · 2021-11-02

Accept (Oral)

---

> ### Author Response · Authors · 2021-12-10
> **Response to all the Reviewers**
>
> We would like to thank all the Reviewers for their encouraging reviews and share their enthusiasm for the results. We respond to all the comments provided by the reviewers below.
>
> Reviewer xKD9 raised the interesting observation that BiT-M, while showing highest correlation with higher brain regions, showed little correlation with the visual voxels. Like the reviewers, we were equally puzzled by this and were interested to understand why this trend was observed.
> We looked at the individual latent vectors or the variance uniquely explained by the BiT-M (in comparison to CLIP and ResNet) using partial correlations. Unfortunately, at least at the moment, we don’t have any obvious hypothesis as to why this model stands out. We also like the idea of looking at other layers in the network and will pick up on the Reviewer’s suggestion in our future work.
>
> Reviewer 3uBV asked for additional information on how the latent representations are built. We have revised the manuscript to make this more clear and hope that the code provided to reproduce the results would further help alleviate any confusion. We would like to thank the reviewer for pointing out the lack of details for the statistical tests and would like to apologise for the oversight. We have added these details to the revised manuscript.
>
> Reviewer 41P3 asked for a few clarifications that we answer below.
>
> (1) All the materials used in the study -- fMRI data and the models -- are publicly available with an accompanying paper discussing them. We cite these relevant papers where these models can be found. To ensure that we use them correctly,  we perform some simple sanity checks (such as the accuracy on the datasets) such that we can reproduce the results mentioned in the corresponding paper. The code we release with this paper has a package that can, in addition to the code provided by the original authors, help to easily load all these models in their correct form.
>
> (2,4) The visual regions correspond to lateral occipital and pericalcarine regions as per the MNI305 atlas. Unfortunately this detail was lost in the paragraph containing various abbreviations, and we have updated the manuscript for better readability.
>
> (3) The trend across all regions was robust to the choice of the different metrics. We provide these plots in the Appendix.
>
> We once again thank all the reviewers for their time and effort.